# Arabidopsis Actin-Binding Protein WLIM2A Links PAMP-Triggered Immunity and Cytoskeleton Organization

**DOI:** 10.3390/ijms252111642

**Published:** 2024-10-30

**Authors:** Prabhu Manickam, Aala A. Abulfaraj, Hanna M. Alhoraibi, Alaguraj Veluchamy, Marilia Almeida-Trapp, Heribert Hirt, Naganand Rayapuram

**Affiliations:** 1BESE Division 4700, King Abdullah University of Science and Technology (KAUST), Thuwal, Makkah 23955, Saudi Arabia; 2Biological Sciences Department, College of Science & Arts, King Abdulaziz University, Rabigh 21911, Saudi Arabia; 3Department of Biochemistry, Faculty of Science, King Abdulaziz University, Jeddah 21551, Saudi Arabia; 4Core Labs, King Abdullah University of Science and Technology (KAUST), Makkah 23955, Saudi Arabia

**Keywords:** plant immunity, stomata, phytohormones, MAPKs, CRISPR-Cas9

## Abstract

*Arabidopsis* LIM proteins are named after the initials of three proteins Lin-11, Isl-1, and MEC-3, which belong to a class of transcription factors that play an important role in the developmental regulation of eukaryotes and are also involved in a variety of life processes, including gene transcription, the construction of the cytoskeleton, signal transduction, and metabolic regulation. Plant LIM proteins have been shown to regulate actin bundling in different cells, but their role in immunity remains elusive. Mitogen-activated protein kinases (MAPKs) are a family of conserved serine/threonine protein kinases that link upstream receptors to their downstream targets. Pathogens produce pathogen-associated molecular patterns (PAMPs) that trigger the activation of MAPK cascades in plants. Recently, we conducted a large-scale phosphoproteomic analysis of PAMP-induced *Arabidopsis* plants to identify putative MAPK targets. One of the identified phospho-proteins was WLIM2A, an Arabidopsis LIM protein. In this study, we investigated the role of WLIM2A in plant immunity. We employed a reverse-genetics approach and generated *wlim2a* knockout lines using CRISPR-Cas9 technology. We also generated complementation and phosphosite-mutated *WLIM2A* expression lines in the *wlim2a* background. The *wlim2a* lines were compromised in their response to *Pseudomonas syringae Pst* DC3000 but showed enhanced resistance to the necrotrophic fungus *Botrytis cinereae*. Transcriptome analyses of *wlim2a* mutants revealed the deregulation of immune hormone biosynthesis and signaling of salicylic acid (SA), jasmonic acid (JA), and ethylene (ET) pathways. The *wlim2a* mutants also exhibited altered stomatal phenotypes. Analysis of plants expressing *WLIM2A* variants of the phospho-dead or phospho-mimicking MAPK phosphorylation site showed opposing stomatal behavior and resistance phenotypes in response to *Pst* DC3000 infection, proving that phosphorylation of WLIM2A plays a crucial role in plant immunity. Overall, these data demonstrate that phosphorylation of WLIM2A by MAPKs regulates *Arabidopsis* responses to plant pathogens.

## 1. Introduction

PAMP-triggered immunity (PTI) refers to plant defense mechanisms triggered by pathogen-associated molecular patterns (PAMPs) [1,2]. During the course of evolution, microbes have developed effector proteins, which are secreted into the host plant via the type III secretion system (T3SS). The PTI pathway is inhibited by various effectors, resulting in effector-triggered susceptibility (ETS) [3]. Plants, in turn, have developed R proteins that recognize pathogen effectors and trigger defensive responses via effector-triggered immunity (ETI) [1,4].

Numerous studies in the last decade have shed light on how mitogen-activated protein kinases (MAPKs) activate PTI during plant–pathogen interactions. MAPKs are well-known for their participation in numerous signaling pathways. Given the various permutations and combinations of MAP3K, MAP2K, and MAPKs involved in the complex MAPK signaling pathway, only a subset of MAPK targets have been discovered so far. We recently carried out a comparative phosphoproteomic analysis in WT and *mapk (mpk3*, *mpk4*, *mpk6)* mutants utilizing flg22 as a microbe-associated molecular pattern (MAMP). The search yielded 152 differentially phosphorylated peptides, 70 of which were identified as potential MAPK targets [5]. Of the 70 targets, At2g39900 (WLIM2A) was selected for further investigation as its role in plant immunity is not well established.

WLIM2A belongs to the family of LIM proteins (LIM proteins are named after the initials of three proteins that are transcriptional factors, namely Lin-11, Isl-1, and MEC-3), which are defined by having one to five LIM domains that associate with other domains, including homeodomains, catalytic domains, cytoskeleton binding domains, and protein binding domains (Src homology 3 (SH3) domains or LD domains—a short amino acid sequence found within proteins that has the consensus sequence LDXLLXXL and functions as a protein-binding interface) [6]; (Appendix A). The LIM proteins are a eukaryotic protein family that is well studied in animals, but not much is known about plant LIM proteins. Recently, it was shown that plant LIM proteins play multiple roles in plant development, metabolism, and defense [7,8,9,10] (Appendix A). Plant LIM proteins are actin-bundling proteins (ABPs), the LIM domains of which interact directly with actin filaments [11]. *Nicotiana tabacum* WLIM1 and *Lily* L1LIM1 have been shown to directly bind to actin filaments and crosslink actin filaments into actin bundles [12,13]. L1LIM1 promotes filamentous actin bundle assembly and protects F-actin filaments against latrunculin B-mediated depolymerization. The appearance of asterisk-shaped F-actin aggregation and the hyper bundle has been associated with defective targeting of endomembrane trafficking and thereby impaired pollen tube elongation. Furthermore, a pleiotropic morphology is observed in pollen tubes overexpressing *L1LIM1*, with retarded pollen development, swollen tip, and multiple tubes emerging out of single pollen grains. These results suggest that L1LIM1 plays an important role in endomembrane trafficking by contributing to actin bundle formation or elongation of pollen tubes [14].

Recently, *Arabidopsis* LIM proteins were identified as RNA-binding proteins (RBPs) in an mRNA interactome study [15]. Several studies suggest that the role of RBPs in RNA transport along the cytoskeleton facilitates mRNA localization [16,17] which suggests that *Arabidopsis* LIMs might also be involved in mRNA transport along actin filaments. In addition, actin function was also studied in the nucleus, where it is involved in mRNA processing, export, and localization [18]. *LIM4* has been shown to bind to actin filaments and bundle them in response to Ca^2+^. Growing evidence points to the importance of the actin cytoskeleton in plant immunity [19]. Actin filament abundance increases exponentially within minutes after receptor activation in *Arabidopsis* epidermal cells [20]. This phenomenon is assumed to represent a unique early feature of PTI responses. The targeted transport of defense chemicals to the infection site, organelle rearrangements, and ligand-induced endocytosis of receptors are all examples of defense responses that require actin remodeling. Furthermore, callose deposition, apoplastic ROS production, and transcriptional reprogramming of defense genes are all significantly reduced when the host actin cytoskeleton is disturbed. Thus, the actin cytoskeleton and associated cellular mechanisms assist the organization of intracellular and apoplastic defenses in host plants. When the actin cytoskeleton is disrupted, both pathogenic and nonpathogenic bacteria are more likely to affect plants. But so far, little is understood about the precise functions of the actin cytoskeleton in host defense.

The actin array in guard cells undergoes dynamic remodeling, which is required for effective stomatal closure. The actin filaments in the closed stomata reorganize from a radial array to a randomly organized network and then to a longitudinal alignment. Disrupting this reorganization via genetic or pharmacological methods results in impaired stomatal closure. The dynamic behaviors of individual actin filaments between guard cells of stomata during the open and closed stages were compared by Li et al. (2019) [21]. It is still uncertain what the first dynamic actin filament events are that may cause the change to the actin structure connected to stomatal closure [22]. Most research on actin organization in guard cells so far has been performed under abiotic stress conditions [23,24]. Shimono et al. (2016) [25] investigated the modifications to actin architecture that take place during the pathogen- and MAMP-induced stomatal closure. Their findings implied that guard cell actin array patterns during immunity are comparable to those during diurnal cycling. Further research is needed to understand the details of actin dynamics during stomatal defense and the underlying molecular pathways.

The main objective of this study was to investigate the role of WLIM2A in plant immunity. We examined the interaction between WLIM2A and three immune MAPKs—MPK3, MPK4, and MPK6—and phosphorylation of WLIM2A by the MAPKs. To understand the function of WLIM2A in plant defense, we analyzed *wlim2a* Arabidopsis knockout mutant lines and assessed their responses to the bacterial pathogen *P. syringae* Pst DC3000 and the necrotrophic fungus *Botrytis cinerea*. Transcriptome analysis was conducted to identify defense-related genes regulated by WLIM2A. Additionally, we investigated the impact of WLIM2A phosphorylation on plant immunity using pathogen assays and examined stomatal closure in wild-type, *wlim2a* mutant, and phospho-mimicking/phospho-dead *WLIM2A* complemented *wlim2A* mutant plants.

## 2. Results

### 2.1. WLIM2A Interacts with and Is Phosphorylated by MAPKs—MPK3, MPK4, and MPK6

WLIM2A (At2g39900) was identified as an MAPK target that is differentially phosphorylated after treatment with flg22. Comparative phosphoproteomics of WT and *mapk* (*mpk3*, *mpk4*, and *mpk6*) mutants suggested the possible involvement of WLIM2A in PTI (Appendix A; [5]). In order to further investigate the role of WLIM2A in MAPK signaling, we first investigated whether WLIM2A can directly interact with any of the three MAPKs, MPK3, MPK4, and MPK6. For this purpose, we tested the interaction between WLIM2A and the three immune MAPKs (MPK3, MPK4, and MPK6) in vivo using bimolecular fluorescence complementation (BiFC) assays. The proteins were transiently co-expressed in leaf epidermal cells of *N. benthamiana.* Confocal images confirmed that WLIM2A strongly interacts with MPK3 in both the nucleus and cytoplasm, while it interacts with MPK4 and MPK6 predominantly in the cytoplasm (Figure 1A).

To further characterize WLIM2A, we tested the capacity of the MAPKs to phosphorylate WLIM2A at the identified T94 in vivo phosphorylation site [5] (Figure 1B,C). We performed in vitro kinase assays with recombinant WLIM2A and constitutively active MPK3, MPK4, and MPK6 proteins. LC-MS/MS revealed that WLIM2A is phosphorylated at T94 by all three MAPKs in vitro, confirming that WLIM2A is a direct MAPK target (Figure 1B). We next examined the subcellular localization of GFP-tagged WLIM2A protein expressed in *N. benthamiana* leaf epidermal cells. Confocal microscopy images showed that WLIM2A localizes to both the cytoplasm and nucleus (Figure 1D). These data aligned with our previous phosphoproteomic study [5], which identified WLIM2A as a potential MAPK target.

### 2.2. Generation of wlim2a Mutants Using the CRISPR-Cas9 System

In order to characterize the function of WLIM2A, we generated mutants using CRISPR-Cas9 technology. Gene-specific primers were designed to amplify the *WLIM2A* fragment from WT Col-0 and CRISPR-Cas9 *wlim2a* lines. We screened for mutations in the coding region of the gene by sequencing. We obtained lines that had either an addition of guanine or thymine in the coding region, which resulted in a frameshift mutation and premature termination of the WLIM2A protein (Figure 2A and Appendix A). To avoid off-target mutations, we selected subsequent generations for the lack of Cas9 by PCR-based genotyping using gene-specific primers (Appendix A). Both *wlim2a-1* and *wlim2a-2* exhibited normal developmental phenotypes and no obvious morphological or developmental defects compared to WT plants (Figure 2B).

### 2.3. Immune Responses to a Virulent Bacterial Pathogen Are Compromised in wlim2a Plants

We investigated the role of WLIM2A in plant immunity upon challenge with *Pst DC3000.* We spray-inoculated *wlim2a-1* and *wlim2a-2* plants with the pathogenic strain *Pst DC3000*. We observed enhanced susceptibility of both of the *wlim2a* mutant lines compared to WT at 72 h post-infection (Figure 3A). We measured the levels of salicylic acid (SA), a key hormone that regulates plant immunity to bacterial pathogens [26]. The data showed significantly lower SA levels in the *wlim2a-1* mutant compared to WT in both untreated and flg22-treated plants (Figure 3B). Therefore, this observation showed that WLIM2A plays a positive role in regulating disease resistance to bacterial pathogen.

As both *wlim2a* mutant alleles exhibited increased susceptibility to *Pst* DC3000, we examined whether the cellular responses involved in PTI were affected in the mutant lines. Increased production of reactive oxygen species (ROS) is one of the earliest immune responses following PAMP perception. Another hallmark of PTI is the activation of various MAPKs, including MPK3, MPK4, and MPK6. *wlim2a-1* mutants showed higher MPK3, 4, and 6 activation levels than WT after 15 and 30 min of flg22 elicitation (Figure 3C). However, flg22-induced ROS production was lower in both *wlim2a-1* and *wlim2a-2* mutants compared to WT plants (Figure 3D).

To study the role of WLIM2A in stress responses, WT plants and *wlim2a-1* and *wlim2a-2* mutants were stained for hydrogen peroxide using diamino-benzidine 3,3′-diamino-benzidine (DAB) and for superoxide radicals using nitroblue tetrazolium (NBT). Polymerization of DAB can be spotted as a brown precipitate in the presence of hydrogen peroxide and NBT as a blue precipitate in the presence of superoxide. Both *wlim2a-1* and *wlim2a-2* mutants accumulated similar H_2_O_2_ (DAB staining) but less O_2_^−^ (NBT staining) compared with WT plants (Figure 3E). We then examined the role of WLIM2A in PAMP-triggered transcriptional responses of known PTI marker genes. Quantitative RT-PCR (qRT-PCR) analysis showed that the basal transcript levels of *FRK1* and *WRK29* were higher in *wlim2a-1* in both untreated and flg22-treated plants when compared with WT plants (Figure 3F).

### 2.4. Global Transcriptomic Profile Shows That WLIM2A Regulates Defense Gene Expression

To characterize the function of WLIM2A in the regulation of gene expression, we carried out a global transcriptome analysis of *wlim2a-1* mutant plants in response to flg22 treatment. The expression levels were normalized using the fragment per kilobase of million mapped reads (FPKM). Differentially expressed genes (DEGs) were identified based on a minimum of a two-fold change in gene expression and false discovery rate (FDR) value (*p* ≤ 0.05). We obtained 1816 DEGs in response to flg22, including 148 upregulated genes and 931 downregulated genes in WT, whereas in *wlim2a-1*, we obtained 554 upregulated and 778 downregulated genes in response to flg22 (Figure 4A and Appendix A).

To gain a better understanding of the genes affected by deletion of WLIM2A, we performed hierarchical clustering analysis of flg22-induced genes followed by GO enrichment analysis. In total, 1816 DEGs were up- and downregulated between WT and *wlim2a-1* treated samples with *p* ≤ 0.05. The clustering analysis of all DEGs was carried out using the Euclidean distance method associated with complete linkage. Ten clusters were identified based on similar expression patterns (Figure 4B). Gene ontology (GO) functional enrichment of DEGs was carried out for the different clusters using AgriGo analysis.

Focusing on three specific clusters of interest, cluster 1 comprised 31 DEGs that were highly upregulated in *wlim2a-1* upon flg22 treatment but not in WT. Enrichment analysis of the GO categories of cluster 1 revealed genes associated with response to localization and macromolecule transport. Cluster 7 and 9 genes were upregulated in untreated *wlim2a-1* plants. Cluster 7 contained 350 DEGs involved in signal transduction, defense response, and cell death, whereas cluster 9 contained 520 DEGs related to the glucosinolate process, response to water and cold, and plastid organization (Figure 4C).

In order to validate the RNA-Seq analysis, we performed qRT-PCR on several of the DEGs. We observed differential upregulation of transcriptional factors *WRKY40* and *AGC2-1* before and after flg22 treatment in *wlim2a-1* compared to WT plants (Figure 4D). The expression of *ERD10* and *PIN7* was repressed after flg22 treatment in *wlim2a-1* compared to WT plants (Figure 4D).

### 2.5. Loss of WLIM2A Function Enhances Basal Defense Against B. cinerea

The transcriptome data revealed many components of phytohormone synthesis and signaling pathways. So, we assessed the concentrations of JA and JA-Ile, two important hormones that control plant immunity. We found no changes in JA, but increased levels of the active jasmonate, JA-Ile, in *wlim2a-1* mutant relative to WT (Figure 5A). To determine the role of WLIM2A in fungal defense, we examined the response of WT and *wlim2a-1* and *wlim2a-2* mutant plants to *B. cinerea*. *wlim2a-1* mutant plants were more resistant than WT (Figure 5B). Higher expression levels of JA-Ile and *B. cinerea*-related genes (such as *BIK1* and *LOX2*) were observed in *wlim2a-1* mutants when compared to WT in flg22-treated plants (Figure 5C). These findings support that WLIM2A plays a negative role in *B. cinerea* defense.

### 2.6. Essential Role of WLIM2A Phosphorylation in Immunity

To study the function of WLIM2A protein phosphorylation by MAPKs at the identified T94 phosphorylation site, we generated stable *WLIM2A-WT* (ProUbi::*WLIM2Awt-GFP)*, *WLIM2A-PM* phospho-mimic (ProUbi::*WLIM2AT94D-GFP)*, and *WLIM2A-PD* phospho-dead (ProUbi::*WLIM2AT94A-GFP)* lines in the *wlim2a-1* mutant background by site-directed mutagenesis. Phenotypically, we found no differences to WT in growth or development of the *WLIM2A-PD* or *WLIM2A-PM* lines (Figure 6B).

We then investigated the role of WLIM2A phosphorylation with respect to defense against the pathogenic bacterial strain *Pst DC3000.* We spray-inoculated the different lines with *PstDC3000* and compared infection levels to WT at 72 h post-infection. In contrast to the enhanced infection levels in *wlim2a-1*, we observed similar susceptibility in *WLIM2A-WT* and *WLIM2A-PM* as in Col-0 (Figure 6A). By contrast, *WLIM2A-PD* had similar *PstDC3000* infection levels as the *wlim2a-1* mutant (Figure 6A). First, these data confirmed that the expression of *WLIM2A-WT* complements the *wlim2a-1* mutant phenotype. Second, ProUbi-driven overexpression of WLIM2AT94D did not confer enhanced *PstDC3000* resistance, and it showed that MAPK-directed phosphorylation of WLIM2A plays a key role in pathogen resistance.

### 2.7. Role of WLIM2A in Stomatal Immunity

Upon pathogen attack, stomata rapidly close as a primary defense mechanism to inhibit pathogen entrance into the leaf apoplast. We observed at 3 hpi increased bacterial titers in *wlim2a-1* and *WLIM2A-PD* mutants compared to WT, *WLIM2A-WT*, and *WLIM2A-PM* lines. Although these differences in bacterial titers were not statistically significant, this trend prompted us to investigate the possible role of WLIM2 in stomatal physiology and stomatal defense (Figure 6A). For this purpose, we treated plants with flg22, which triggered stomatal closure, or with H_2_O as mock control. Under mock treatment, we observed differences in stomatal opening of WT, *wlim2a-1*, *wlim2a-2*, *WLIM2A-WT*, *WLIM2A-PD*, and *WLIM2A-PM*. Upon flg22 treatment, stomata of WT, *WLIM2A-WT*, and *WLIM2A-PM* closed in response to the treatment, but not in the *wlim2a-1* and *WLIM2A-PD* lines (Figure 6B–D). Together, these findings indicated that WLIM2A is crucial for the stomatal immunity response in *Arabidopsis*.

## 3. Discussion

To protect themselves from a wide range of biotic threats, plants have developed numerous intricate immune response mechanisms that help them withstand these challenging conditions. Plant MAPKs play an important role in response to several biotic and abiotic stresses [27] by phosphorylating their substrates, ranging from transcription factors to chromatin-associated proteins and several other cytosolic resident proteins. But so far, only a fraction of the MAPK substrates have been identified. MAPK-dependent phosphorylation of their substrates and their subsequent functional characterization is vital to plant stress responses. WLIM2A was identified from phosphoproteomic screening and phosphorylated in vivo by MPK3, MPK4, and MPK4 at T94 in response to flg22 treatment [5]. WLIM2A is an actin-binding protein. The role of actin-binding proteins on plant cytoskeleton dynamics during pathogen infection and the activation of immunity has been well documented [21,25]. In this study, we undertook a thorough genetic and phytopathology-based approach to investigate the role of WLIM2A in regulating stomatal immunity and bacterial infection.

Protein interaction by BIFC and in vitro phosphorylation assays clearly confirmed that WLIM2A interacts with and is phosphorylated by the MAPKs at T94, as identified in the *in vivo* phosphoproteomic screening. Based on all the published microarray gene expression data from AtGenExpress, WLIM2A is expressed in all organs and developmental stages (Appendix A). Due to the lack of good T-DNA insertion mutants, we generated CRISPR-cas9 mutants. The mutants of WLIM2A did not exhibit any morphological or developmental defects (Figure 2B). However, when challenged with pathogens, *wlim2a* mutants showed that that they are more susceptible to the bacterial pathogen *Pst DC3000* and more resistant to the necrotrophic fungal pathogen *B. cinerea*. Furthermore, *wlim2a* mutant plants displayed deregulation in the production of ROS, reduced levels of the phytohormone SA, activation of MAPKs, and expression of pathogenesis-related genes such as *FRK1* and *WRKY29* following PAMP treatment. Overall, these findings suggest that WLIM2A is essential for plant immunity.

It has been reported that WLIM2A regulates gene transcription in the nucleus [28]. We analyzed the transcriptional changes following flg22 treatment using RNA-Seq. We found that genes encoding WRKY40, a regulator of SA synthesis, NPR3, a receptor of SA immune signaling, and CBP60G, another regulator of SA, were deregulated in the *wlim2a* mutant after flg22 treatment. Defense-related genes, including *WRKY27*, *PEN2*, *WRKY40*, *MYB51*, and *ERF6*, were upregulated in the flg22-treated WT and *wlim2a-1* mutant when compared to mock-treated plants. MPK3/MPK6 phosphorylate ERF6 and thereby promote the biosynthesis of indole glucosinolates. MPK3 and MPK6 regulate the function of MYB51, a key regulator of IGS biosynthesis, through ETHYLENE RESPONSE FACTOR6 (ERF6), which is a target of PENETRATION2 (PEN2)-dependent chemical defense in plant immunity [29]. MPK3 and MPK6 also have an essential function in the induction of camalexin, a major phytoalexin in *Arabidopsis* [30]. Moreover, *Botrytis cinerea* infection-activated MPK3/MPK6 promotes the biosynthesis of indole-3yl-methylglucosinolate (I3G), which is converted into 4 methoxindole-3-yl-nethylglucosinolate (4MI3G). These results support the idea that WLIM2A might control glucosinolate biosynthesis and is involved in plant immunity, although further validation and studies are required. The expression of *PIN7* and *LOX2* was significantly increased in the *Arabidopsis wlim2a* mutant, whereas these genes were downregulated after flg22 treatment in the *wlim2a* mutant and WT. JA and auxin may interact positively in plant resistance to the necrotrophic pathogen *B. cinerea* [31]. PIN7 is an auxin transporter, and LOX2 is involved in JA biosynthesis. JA-dependent defense signaling could be part of the auxin-mediated defense mechanism involved in resistance to necrotrophic pathogens [32]. These results suggest that the altered susceptibility phenotype of *wlim2a* mutants to infection by bacterial and fungal pathogens involved the transcriptional regulation of defense-related genes.

Our earlier phosphoproteomic analysis indicated that T94 phosphorylation of WLIM2A was dependent on MAPKs. We showed that MAPK phosphorylation at T94 of WLIM2A is indeed dependent on MAPKs. We further generated mutations at T94 to either mimic a constitutively phosphorylated or a phospho-deadWLIM2A version and investigate the effects of phosphorylation on immunity responses. When challenged with Pst DC3000, the phospho-mimic line behaved like WT, whereas the phospho-dead line behaved like the *wlim2a* mutant, indicating that phosphorylation of WLIM2A T94 is essential to regulate defense. Furthermore, when we observed the stomatal physiology in response to flg22 treatment, the phospho-dead line behaved similar to the *wlim2a* mutant, indicating that the phosphorylation of WLIM2A is crucial for stomatal immunity in Arabidopsis.

Earlier studies of the SA-deficient *nahG* mutant and the SA-biosynthetic *sid2/eds16* mutants clearly demonstrated that SA biosynthesis is essential for flg22-induced stomatal closure. *wlim2a* mutant plants exhibited decreased SA accumulation. Furthermore, SA was found to promote stomatal closure via peroxidase-mediated extracellular ROS production. *wlim2a* mutants showed decreased ROS production following flg22 treatment. Our findings suggest that the immune defects of *wlim2a* mutants, including stomatal immunity might be associated with a hormone imbalance in salicylic acid.

Actin-binding proteins (ABPs) are essential for organizing actin filaments in plants. In Arabidopsis, over 70 ABPs are involved, with four main families—fimbrin, formin, villin, and LIM domain-containing proteins (LIM)—being critical for actin bundle formation and maintenance (Appendix A). Research by Papuga et al. (2010) [11] revealed that WLIM2A, a member of the LIM family, directly interacts with actin filaments, stabilizing them and promoting higher-order structures like actin bundles. This interaction was confirmed through various biochemical assays, such as co-sedimentation and depolymerization, which demonstrated the role of WLIM2A in stabilizing actin filaments. While the role of LIM proteins in actin dynamics is relatively well understood, their function in plant immunity remains underexplored. Actin cytoskeleton remodeling is essential for various immune responses, such as transcriptional activation, reactive oxygen species (ROS) generation, and vesicle trafficking. Studies indicate that disrupting the actin cytoskeleton can enhance pathogen resistance in plants through a salicylic acid-dependent mechanism [33,34]. Specifically, microfilament (MF) disruption appears to increase resistance to bacterial pathogens, while microtubule (MT) disruption enhances susceptibility. However, the opposite occurs in fungal infections, where the inhibition of both MF and MT reduces resistance [35].

Large-scale phosphoproteomic analyses revealed that WLIM2A is phosphorylated in vivo at T94 by MPK3, MPK4, and MPK6 [5]. Earlier studies showed that LIM proteins can crosslink cytoplasmic actin filaments and act as transcription factors in the nucleus [36]. This phosphorylation may regulate the subcellular localization of LIM proteins, shuttling them between the cytoplasm and nucleus to activate gene transcription.

Perception of microbe-associated molecular patterns (MAMPs) by plant surface receptors triggers rapid signaling events, which affect the actin cytoskeleton by regulating ABPs. Failure to rearrange actin filaments compromises stomatal closure, a critical defense response [37]. LIM proteins like AtLIMs stabilize actin bundling during these immune responses [38]. In particular, the MPK3/6 pathway is crucial for disassembling the actin array to close stomata during pathogen attack [29]. Our studies show that WLIM2A is rapidly phosphorylated upon flg22 activation, initiating stomatal immunity. Further research should investigate the precise role of WLIM2A phosphorylation in actin dynamics during plant defense (Figure 7).

## 4. Materials and Methods

### 4.1. Plant Materials and Growth Conditions

*Arabidopsis thaliana* ecotype Columbia-0 (Col-0) was used as wild-type (WT) plants. Seeds were surface-sterilized and stratified at 4 °C on growth medium for at least two days. The growth medium was composed of ½ Murashige and Skoog basal salts with minimal organics, 0.05% MES hydrate, and 0.5% agar type (Sigma A4675, St. Louis, MO, USA). The pH was adjusted to 5.7 with KOH. Seedlings were grown at 23 °C day/22 °C night in long day conditions, i.e., 16 h light/8 h dark photoperiod, for the indicated time. For cultures in Jiffy pots in Percival growth chambers, seeds were stratified at 4 °C in water for at least two days, and then grown at 23 °C day/22 °C night, 60% humidity in short day conditions, i.e., 8h light/16 h dark photoperiod for four weeks.

*Nicotiana benthamiana* (*N. benthamiana*) plants were grown in the greenhouse at 28 °C under 70% humidity and 16 h/8 h light conditions. Four-week-old plants were used for transient leaf transformation by agroinfiltration to carry out BiFC and subcellular localization assays.

### 4.2. Generation of wlim2a Mutant Lines Using CRISPR-Cas9 Technology

To generate WLIM2A knockout mutant Arabidopsis lines, we designed 20 nucleotide-long guide RNAs (gRNAs) targeting the WLIM2A gene using the CRISPR-P 1.0 method [39]. Two gRNA targets with the lowest off-target scores were selected. The gRNA cassettes were cloned into a mcherry pHSE401 vector containing an egg cell-specific promoter, as previously described [13]. These constructs were then transformed into Arabidopsis Col-0 plants using the agrobacterium mediated-floral dip method.

Seeds collected from the T0 plants were screened on ½ MS media plates supplemented with hygromycin to identify resistant seedlings, which were subsequently transferred to soil. Genomic DNA was extracted from the T1 transgenic lines for further analysis. To assess mutations in WLIM2A, DNA fragments surrounding the gRNA target sites were amplified by PCR using specific primers. The PCR products were purified and sequenced with WLIM2A-specific primers. The sequencing data were analyzed using CLC Workbench and CRISPR-Cas9-induced insertions/deletions were identified (Appendix A).

Counterselection PCR with Cas9-specific primers was performed to screen for non-transgenic T2 plants, ensuring that the desired mutations were not linked to the transgene (Appendix A). This permitted us to identify WLIM2A knockout mutant Arabidopsis lines.

### 4.3. Quantitative Real-Time PCR

For PAMP-induced defense gene expression, 14-day-old *Arabidopsis* WT and mutant seedlings grown on ½ MS agar were transferred onto liquid ½ MS and cultured overnight before flagellin 22 (flg22) treatment. The flg22 peptide was dissolved in double-distilled H_2_O to obtain a stock solution of 1 mM, then further diluted to inoculate at a concentration of 1 µM. Seedlings were treated with either 1 µM flg22 or water (mock) for 1 h. WT and mutant seedlings samples (100 mg) were homogenized in liquid nitrogen. Total RNA was extracted using NucleoSpin Plant RNA kits, cDNA was reverse-transcribed from 1 µg of total RNA using SuperScript III First-Strand Synthesis Kit (Invitrogen, Vilnius, Lithuania), and the transcript levels of genes were assessed by qRT-PCR using the PTI marker gene primers (Appendix A). Quantitative RT-PCR was performed using SsoAdvanced Universal SYBR Green Supermix (Bio-Rad, Carlsbad, CA, USA). The data were analyzed using Bio-Rad CFX manager software Version. 3.0. At3g18780 (Actin) and At4g05320 (UBQ10) were used as reference genes for normalization of gene expression levels in all samples, then the normalized gene expression levels were expressed relative to wild-type controls in each experiment.

### 4.4. Pathogen Assays

WT and *wlim2a* Arabidopsis plants were grown for four weeks under short day conditions and spray-inoculated with *Pseudomonas syringae pv.tomato* DC3000) (*Pst* DC3000) at OD_600_ = 0.2 with 0.04% (*v*/*v*) Silwet L-77. Disease symptoms were evaluated at 3 and 72 h post-infection; leaf discs were sampled and bacteria were extracted. In total, three biological replicates were taken from 30 plants of each plant genotype. Bacteria were extracted with 10 mM MgCl_2_ containing 0.01% (*v*/*v*) Silwet L-77. The homogenates were plated on LB agar media containing rifampicin (50 mg/L) after four ten-fold serial dilutions, incubated at 28 °C for 48 h, and bacterial colonies were counted. The infection level of the leaf samples from three biological replicates was reported as CFU/cm^2^.

For *Botrytis cinerea* infection, WT and *wlim2a* Arabidopsis plants were grown under short day conditions for four weeks and inoculated by placing a 5 µL droplet of a fungal spore suspension (5 × 10^6^ spores/mL) on each rosette leaf. Disease symptoms were evaluated after 72 h post-infection, pictures of the plant leaves were taken and quantified using ImageJ Software Version 1.53. In total, three biological replicates of 9 plants each per plant genotype were evaluated.

### 4.5. ROS Burst Assays

ROS burst assays were carried out using the luminol-based luminescence method [26]. Briefly, 4 mm leaf discs of four weeks old *Arabidopsis* WT and mutant plants were incubated overnight (adaxial side up) in 150 µL of sterile water in 96-well plates (Thermo Fisher, Rochester, NY, USA). The next day, the water was replaced with 100 µL of reaction solution containing 50 µM of luminol (Sigma, St, Louis, MO, USA), 10 µg/mL of horseradish peroxidase (Sigma, St, Louis, MO, USA), and supplemented with 1 µM of flg22 or water as a mock control. Luminescence was recorded at 1 min intervals after the addition of flg22 for 40 min using a TECAN infinite 200 PRO microplate reader. The signal integration time was 0.5 s and the ROS measurements were expressed as means of RLU (relative light units).

### 4.6. DAB and NBT Stainings

Four-week-old *Arabidopsis* WT and mutant plant leaves were stained for superoxide radicals and hydrogen peroxide using nitroblue tetrazolium (NBT) (N6876, Sigma-Aldrich, Manheim, Germany) and 3,3′diaminobenzidine (DAB) (D5637, Sigma-Aldrich, St. Louis, MO, USA), respectively. The leaves were then mounted on slides with 50% glycerol and photographs taken using a Nikon SMZ25 stereomicroscope.

### 4.7. pTEpY Assays and Immunoblotting

Total proteins were extracted from *Arabidopsis* WT and *wlim2a* seedlings after growth on ½ MS plates for 14 days. The seedlings were frozen in liquid nitrogen and powdered using a tissue lyser, resuspended in 200 µL of SDS sample buffer, heated at 85 °C for 10 min, centrifuged at 18,000× *g* for 10 min at 4 °C, and the supernatant was collected. Using 10% SDS-polyacrylamide gels, total proteins were resolved and electro-transferred onto PVDF membranes (BIO-RAD). The blots were blocked for one hour at room temperature with 5% (*w*/*v*) BSA in TBST before being incubated overnight at 4 °C with anti-phopho-p44/42 ERK antibody (diluted 1/5000) in 2% (*w*/*v*) BSA in TBST (#4370, Cell Signaling, Danvers, MA, USA). HRP-conjugated goat anti-rabbit IgG (Promega, Madison, WI, USA) was used as the secondary antibody. HRP activity was detected with a chemiluminescent reagent. The immunoblots was stained with Ponceau-S for protein visualization. Each immunoblotting analysis shown is representative of three independent biological repeats.

### 4.8. RNA Sequencing Analysis

The 14-day-old *Arabidopsis* WT and mutant seedlings were grown in three independent biological replicates on ½ MS agar, moved to ½ MS liquid media overnight, and then treated for one hour with 1 µM flg22 peptide (GeneScript) or water (mock). The seedlings were collected for RNA-Seq and 1 µg of total RNA was used for RNA-Seq library preparation using the Illumina Truseq Stranded mRNA Sample Preparation LS (low sample) kit, following the manufacturer’s protocol. Three biological replicates were analyzed for each condition. The sequencing was performed on the HiSeq 4000 platform with a read length of 300 bp paired ends. Reads were quality-controlled using FASTQC v0.11.5 on 26 January 2021 (http://www.bioinformatics.babraham.ac.uk/projects/fastqc/). Trimmometric was used for quality trimming. The parameters for read quality filter were set as follows: minimum length of 36 bp; mean Phred quality score greater than 30; leading and trailing base removal with a base quality below 3; and sliding window of 4:15. TopHat v2.1.1 was used to align short reads to the *Arabidopsis thaliana* TAIR 10 reference genome, with Cufflinks used for transcripts assembly. The differentially expressed genes (DEGs) (fragment per kilobase of transcript per million mapped reads, FPKM) were identified using Cufflinks and the limma package in R Version 3.19. Cuffdiff v2.2.1 [40] was used with quartile normalization to find significant gene expression. Genes with 2-fold change and *p*-value ≤ 0.05 were considered as significantly different between samples with and without flg22 treatment. Hierarchical clustering of these genes was performed using Mev v4.8.1 [41]. Gene ontology (GO) classification was performed with the AgriGo [42] and DAVID Bioinformatic software v2022q1 (https://david.ncifcrf.gov/). Venn diagrams were generated using (http://bioinfogp.cnb.csic.es/tools/venny/).

### 4.9. Phytohormone Analysis

*Arabidopsis* WT and mutant seedlings were grown for two weeks on ½ MS agar, then moved to ½ MS liquid media overnight. The seedlings were then treated for an hour on ½ MS with 1 µM flg22 peptide or water (mock). Phytohormone extraction was carried out as previously reported [43]. A Thermo Fisher TQS-Altis Triple Quadrupole Mass Spectrometer linked to a Thermo Scientific Vanquish MD HPLC system was used to quantify the chemicals using HPLC-ESI-SRM. The compounds were eluted using water (A) and acetonitrile (B) as mobile phases at 0.6 mL/min and in gradient elution mode, as follows: 10% B for 0.5 min, 10–55% B at 4.5 min, 55–100% B at 4.7 min, and 100% until 6.0 min. The temperature in the column was set to 55 °C. To determine the statistical significance of three replicates, ANOVA was used, followed by the Tukey test.

### 4.10. Site-Directed Mutagenesis and Generation of Phospho-Mimick and Phospho-Dead Mutants

Wild-type sequences of *WLIM2A* were amplified by PCR from *Arabidopsis* cDNA and cloned into PENTR-D-TOPO vector (Appendix A). Mutated versions of *WLIM2A* were generated by site-directed mutagenesis using the Gene Art Site-Directed Mutagenesis System kit from Thermo Fisher Scientific. The primers designed to introduce the mutations are listed in Appendix A. The mutations were confirmed using Sanger sequencing (Appendix A) and the mutated versions were transformed into the final Gateway destination vector using LR clonase (Life technologies, Thermo Fisher Scientific, Carlsbad, CA, USA). The constructs were transformed into *Agrobacterium* cells and introduced into the homozygous *wlim2a-1* and *wlim2a-2* lines using the agrobacterium-mediated floral dip method.

### 4.11. Bimolecular Fluorescence Complementation (BiFC)

To generate expression plasmids, the coding sequences of At2g39900 and MAPKs were cloned into the N- or C-terminal part of YFPs under the promoter of 35S::GFP in the pBIFC1, 2, 3, and 4 vectors. A range of positive and negative controls was prepared for the experiments. The constructs were transformed into the *Agrobacterium tumefaciens* C58C1 strain. The cultures were grown on LB medium with the corresponding antibiotic selection marker for 24 h at 28 °C. The culture was centrifuged and resuspended in infiltration buffer (10 mM MgCl_2_, 10 mM MES pH 5.7, 150 µM acetosyringone) to an OD600 of 1.5 and kept in the dark for 3 h. The P19 viral suppressor of gene silencing was used to co-express with each combination to prevent silencing of transiently expressed proteins. A bacterial culture of each combination was mixed before infiltration. For transient expression, the resultant bacterial suspension was directly infiltrated into *N. benthamiana* leaves. Three days after infiltration, the leaves were visualized using an upright LSM 710 Zeiss confocal microscope with a 20× objective (Plan-Apochromat, NA1.0, White Plains, NY, USA) for YFP fluorescence. All images were obtained using the Argon laser with 514-nm excitation.

### 4.12. Subcellular Localization Studies

The coding sequence of At2g39900 was cloned into p35S::C-GFP in the pGWB5 vector (Appendix A). The construct was transformed into the *Agrobacterium tumefaciens* C58C1 strain. For transient expression, the resultant bacterial suspension was directly infiltrated onto *N. benthamiana* leaves. An upright Zeiss LSM710 confocal microscope fitted with a 20× objective (Plan-Apochromat, NA1.0, White Plains, NY, USA) was used to visualize the leaves with GFP fluorescence three days after infiltration. All images were acquired using an argon laser at 514 nm.

### 4.13. In Vitro Kinase Assays

The purified recombinant WLIM2A protein and constitutively active MAPKs were mixed together in kinase reaction buffer. The kinase buffer (20 mM Tris-HCl pH 7.5, 10 mM MgCl_2_, 5 mM EGTA, 1 mM DTT, and 50 µM ATP) and the reaction mixtures were incubated at ambient temperature for 30 min. SDS sample buffer was added to stop the reaction and the samples were denatured by heating at 95 °C for 10 min. The protein samples were resolved using SDS-PAGE. SimplyBlue^TM^ SafeStain (Novex cat. no. LC6065, Carlsbad, CA, USA) was used to stain the gel, and the band corresponding to the protein of interest was excised, cut into small pieces of 0.5 mm^3^, and de-stained with four successive washes of 15 min each with ACN and 100 mm NH_4_HCO_3_. Proteins were reduced at 37 °C for 1 h with 10 mm Tris(2-carboxyethyl) phosphine (TCEP, C-4706 Sigma) in 100 mm NH_4_HCO_3_, followed by alkylation at ambient temperature for 30 min with 20 mm S-methyl methanethiosulfonate (MMTS, 64306 Sigma, St. Louis, MO, USA). Proteins were then digested overnight at 37 °C with trypsin (porcine trypsin, Promega, Fitchburg, WI, USA). After stopping the digestion with 1% formic acid, the peptides were recovered by incubating the gel pieces in ACN. The desalted peptide solution was analyzed by LC-MS/MS using a C18 ZipTip^®^ (Millipore, Burlington, MA, USA, Cat. No. ZTC18S096).

## Figures and Tables

**Figure 1 ijms-25-11642-f001:**
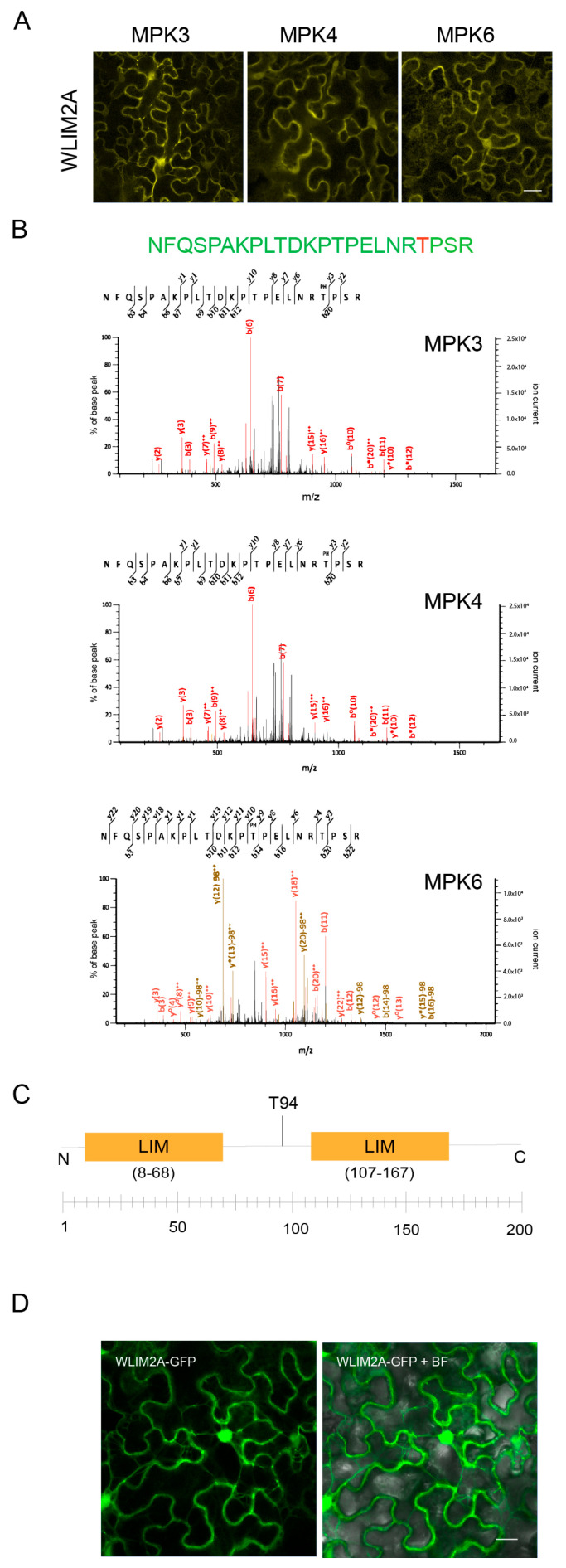
WLIM2A is a phospho-protein. (**A**) BiFC investigation using MPK3, MPK4, and MPK6 in *N. benthamiana* leaf epidermal cells with the MAPK candidate substrate WLIM2A. Laser scanning confocal microscopy was used to detect YFP fluorescence. Scale bar = 20 µM. (**B**) Analysis of samples from an *in vitro* kinase test with MPK3, MPK4, and MPK6 kinases and WLIM2A fragment by mass spectrometry. Mass spectra for the peptide are shown, *b* and *y* ions are indicated on the spectral lines. (**C**) WLIM2A protein structure shows the relative position of two LIM domains and position of the phospho-site obtained from the phosphoproteomic screening. (**D**) Subcellular localization of WLIM2A-GFP transiently expressed in *N. benthamiana* cells under the 35S promoter. The tagged proteins were expressed in 4-week-old tobacco plants, and the localization was observed using laser scanning confocal microscopy between 48 and 72 h after infiltration. Scale bar = 50 µM.

**Figure 2 ijms-25-11642-f002:**
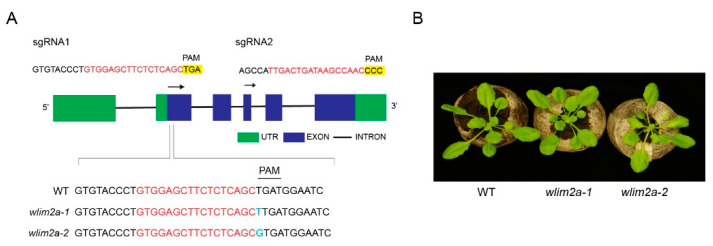
*wlim2a* mutant lines. (**A**) A schematic representation of the CRISPR-Cas9 genome-editing strategy in *Arabidopsis thaliana* to target *WLIM2A*. Based on two sgRNAs targeting *WLIM2A* and sequencing reads from selected mutant alleles, T_0_ transformants show the addition of thymine or guanine. (**B**) WT (Col-0), *wlim2a-1*, and *wlim2a-2* mutants have the same morphological phenotype. Plants cultivated in Jiffy pellets for four weeks are shown.

**Figure 3 ijms-25-11642-f003:**
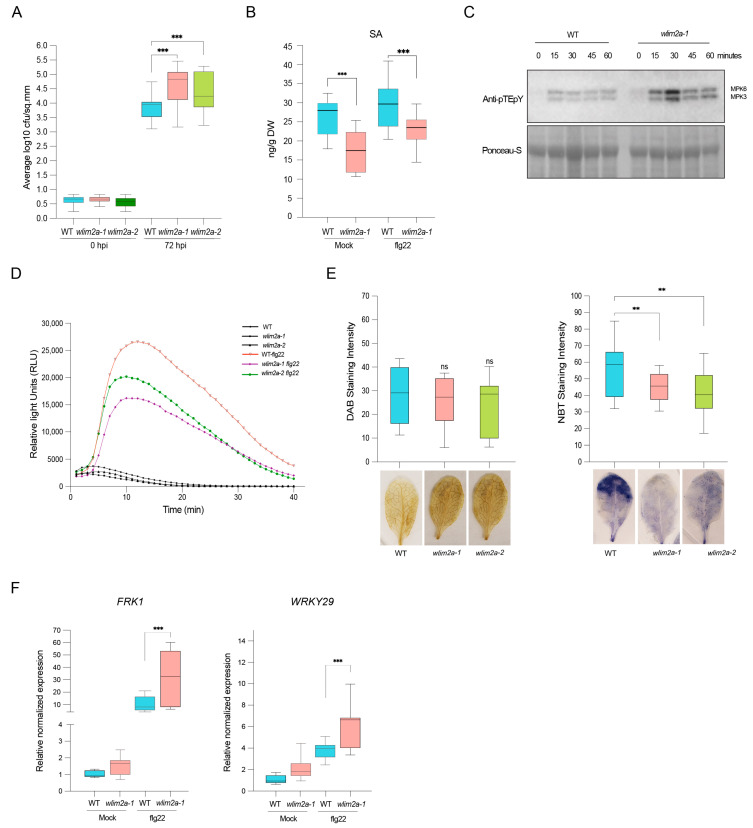
*wlim2a* mutant plants are susceptible to bacterial infection. (**A**) *Arabidopsis* plants *wlim2a-1*, *wlim2a-2*, and WT (Col-0) infected with *Pst* DC3000. Four-week-old seedlings were spray-inoculated with Pst DC3000 or mock. Pst DC3000 colonies were counted at Day 0 and Day 3 after infection. The vertical bars represent the standard error for the three biological replicates with 16 plants each. Student’s *t*-test, *p* < 0.05, was used to determine statistical significance. The median is shown by the lines in the boxes. (**B**) SA levels were quantified by LC-MS/MS in Col-0, *wlim2a-1*, and *wlim2a -2* lines after treatment with or without flg22 for 1-h. Eighteen plants for each line were used in three biological replicates with 3 seedlings at 100 mg/tube each. (**C**) MPK3 and MPK6 are rapidly and transiently activated by flg22 in mock and *wlim2a-1* mutants. Col-0 and the *wlim2a-1* mutant were incubated in water overnight before being exposed to 1 µM flg22 for the specified amount of time. By utilizing an anti-pTE-pY antibody against the phosphorylated version of ERK2, western blotting was used to detect the phosphorylation of MPK3 and MPK6. Equal loading of proteins was shown by Ponceau-S staining. (**D**) In *wlim2a* mutants, flg22-induced ROS generation is reduced. Leaf disks from four-week-old *Arabidopsis thaliana* Col-0, *wlim2a-1*, and *wlim2a-2* plants were collected and treated with 1 µM flg22 or mock. Over 40 min, a luminol-based assay detected ROS accumulation as relative light units (RLU). Three independent experiments generated similar findings (*n* = 12/treatment). (**E**) Evaluation of H_2_O_2_ levels in Col-0 and mutants using 3,3′-diaminobenzidine staining (DAB) and O_2_^−^ levels in mutants using nitroblue tetrozolium (NBT) staining in comparison to WT Col-0. (**F**) PTI marker gene relative expression levels in WT (Col-0), *wlim2a-1*, and *wlim2a-2*. *UBIQUITIN* and *ACTIN* were used as reference genes to normalize the data. The X-axis displays the expression of *FRK1* and *WRKY29* in mock- and flg22-treated plants. The Y-axis displays the changes in gene expression levels shown on a log 2 scale. Three independent experiments were carried out, each with *n* = 9. The standard error (SE) of the data is shown by the error bars (** denotes *p* < 0.01 *** denotes *p* > 0.001, ns denotes Not Significant, Student’s *t*-test).

**Figure 4 ijms-25-11642-f004:**
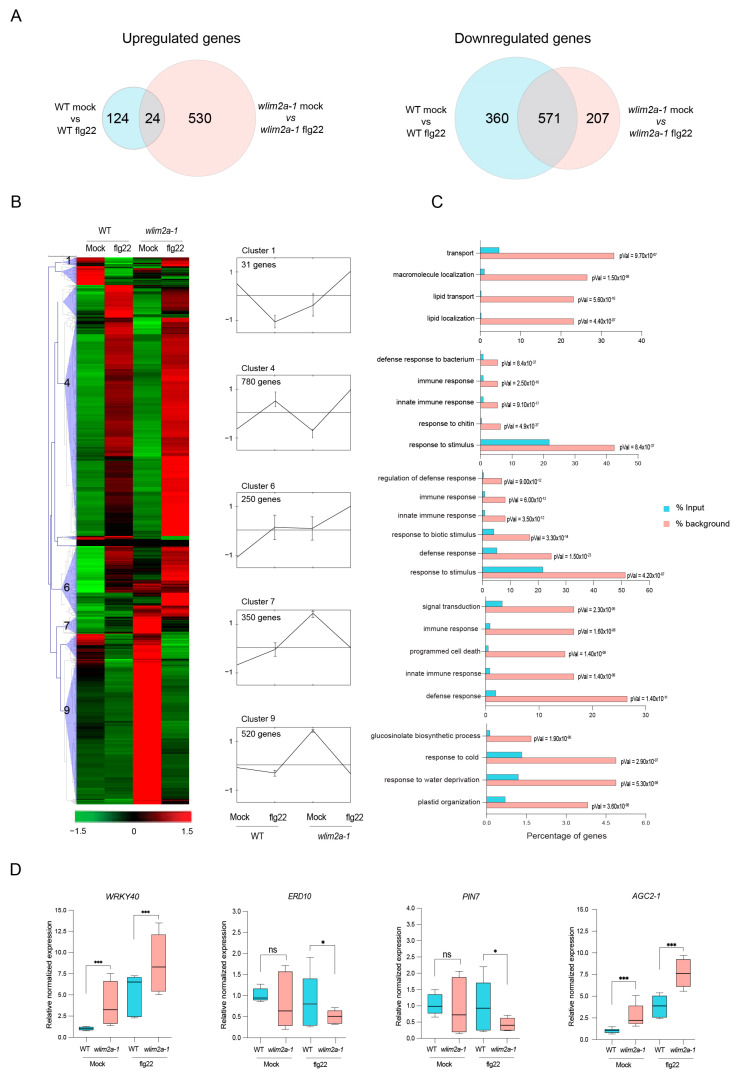
Transcriptomic analyses in WT and *wlim2a* mutant plants. (**A**) Venn diagram displaying the overlap of differentially expressed genes (DEGs) identified by RNA-Seq between WT mock vs. WT-flg22 and *wlim2a-1* mock vs. *wlim2a-1* flg22. (**B**) The DEGs in WT mock, WT-flg22, *wlim2a-1* mock, and *wlim2a-1* flg22 were clustered hierarchically into ten groups. The values are provided as log 2-fold changes (*p* < 0.01). In the clustering, the average linkage method and Pearson correlation (MEV4.0) were applied. The red and green colors represent up- and downregulated expression, respectively. The black lines reflect the average relative expression levels, whereas the gray lines represent the relative expression of each gene in clusters 1, 4, 6, 7, and 9. (**C**) Functional enrichment of DEGs between *Arabidopsis* Col-0 (WT) mock, *wlim2a-1* mock, WT flg22, and *wlim2a-1* flg22 in the GO databases. Blue represents input and orange represents the reference control. (**D**) RT-qPCR validation of RNA-Seq data. *ACTIN* and *UBIQUITIN* were used to standardize the signal intensity for each transcript. Gene expression in WT mock, WT flg22-treated, *wlim2a-1* mock, and *wlim2a-1* flg22-treated is shown on the X-axis. The Y-axis displays the log 2 scaled variations in gene expression levels. The error bars represent SE (standard error), one asterisk (*) indicates *p* < 0.05, three asterisks (***) indicate *p* > 0.001, ns denotes Not Significant, Student’s *t*-test.

**Figure 5 ijms-25-11642-f005:**
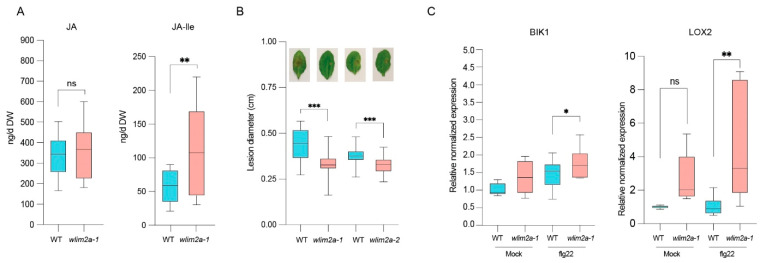
*wlim2a* mutants have increased JA-Ile levels and increased resistance to *B. cinerea*. (**A**) Quantitative study of JA and JA-lle levels in Col-0, *wlim2a-1*, and *wlim2a-2* lines using LC MS/MS with or without flg22 treatment for 1 h. Plants of three seedlings of 100 mg/tube and 18 plants for each line with three biological replicates. (**B**) *Arabidopsis wlim2a* mutants exhibited enhanced resistance to the fungus *B. cinerea*. Four-week-old *Arabidopsis* plants were inoculated with 5 μL droplets of fungal spores (5 × 10^5^ spores mL^−1^). The area of lesion in the leaves of Col-0 and *wlim2a* mutant plants was measured 48 h later. ImageJ software Version 1.53 was used to determine the size of the lesions. Vertical bars represent the standard error for three biologically independent experiments (*n* = 8 for each). Three leaves per plant, with an average of 8 plants per replicate. (**C**) *BIK1* and *LOX2* gene expression in WT mock, WT flg22-treated, *wlim2a-1* mock, and *wlim2a-1* flg22-treated *Arabidopsis* plants. The error bars represent SE (standard error), one asterisk (*) indicates *p* < 0.05, two asterisk (**) indicates *p* < 0.01, three asterisks (***) indicate *p* > 0.001, ns denotes Not Significant, Student’s *t*-test.

**Figure 6 ijms-25-11642-f006:**
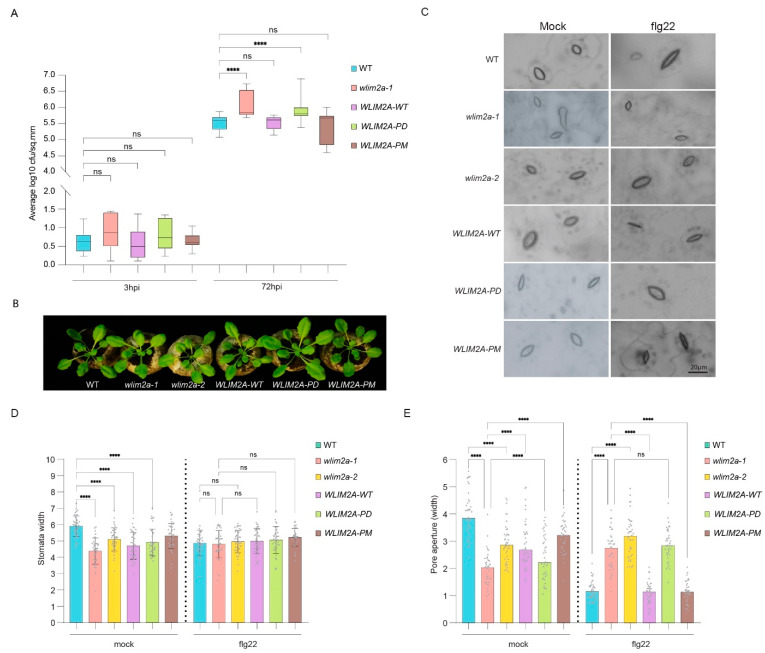
WLIM2A plays an essential role in stomatal immunity in *Arabidopsis*. (**A**) Four-week-old Arabidopsis seedlings of *wlim2a-1*, *wlim2a-2*, *WLIM2A*-*WT*, *WLIM2A*-*PD*, *WLIM2A*-*PM*, and WT (Col-0) were spray-inoculated with *Pst* DC3000 bacterial suspension or mock. The colonies were counted 3 and 72 h after inoculation (hpi). The vertical bars represent the standard error for the three biological replicates with 16 plants each. (B) Representative plant photos of WT, *wlim2a-1*, *wlim2a-2*, *WLIM2A*-*WT*, *WLIM2A*-PD, and *WLIM2A*-*PM* after 72 h of exposure to *Pst* DC3000. (**C**) Representative images of the stomatal closure in the MAMP-induced stomatal defense in WT, *wlim2a-1*, and *wlim2a-2*, *WLIM2A*-*WT*, *WLIM2A*-*PD*, and *WLIM2A*-*PM*. (**D**) Width of the Stomatal guard cells of WT, *wlim2a-1*, *wlim2a-2*, *WLIM2A*-*WT*, *WLIM2A*-PD, and *WLIM2A*-*PM*. (**E**) Stomatal aperture measurement of WT, *wlim2a-1*, *wlim2a-2*, *WLIM2A*-*WT*, *WLIM2A*-PD, and *WLIM2A*-*PM*. Scale bar = 20 µM. The error bars represent SE (standard error), Four asterisks (****) indicate *p* > 0.001, as determined by one-way ANOVA with Tukey’s multiple comparison test, ns denotes Not Significant, Student’s *t*-test.

**Figure 7 ijms-25-11642-f007:**
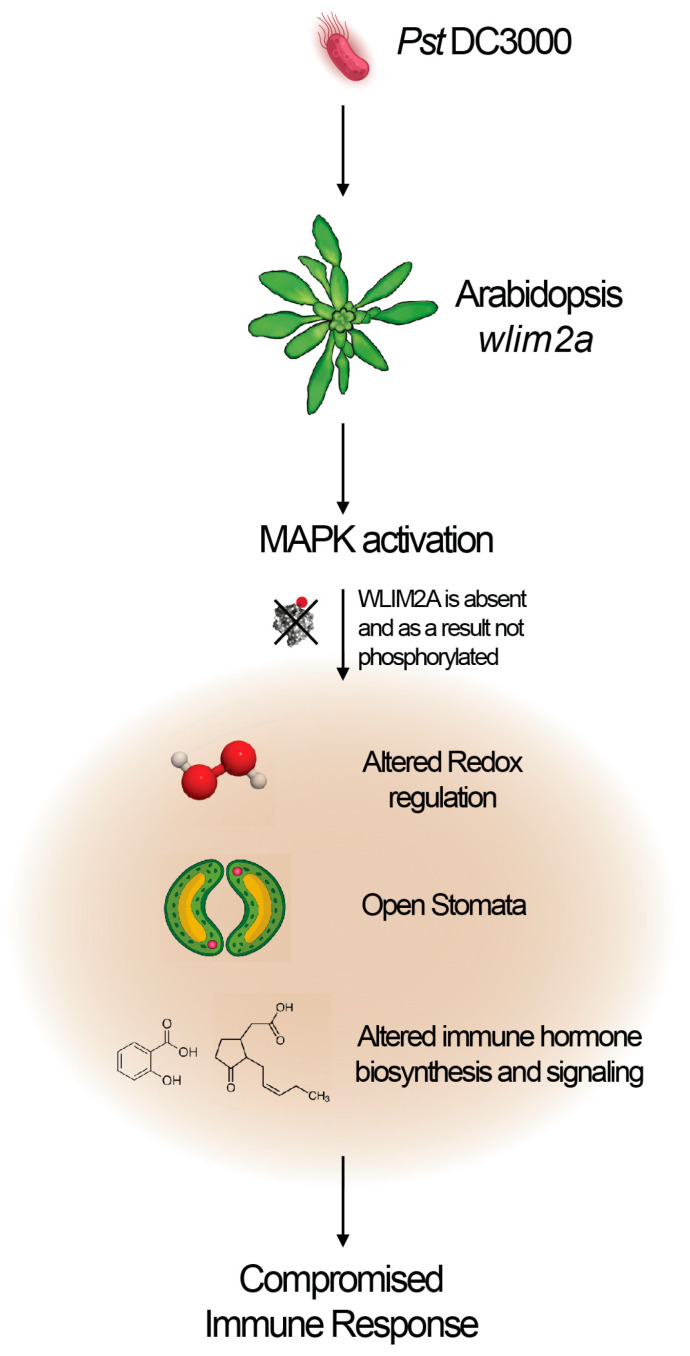
A working model for the role of WLIM2A(LIM2) in plant immunity. In the *wlim2A* mutant, upon flg22 perception, WLIM2A is absent and as a result is not phosphorylated by MAPKs. Upon pathogen infection, *wlim2a* plants exhibit altered redox changes, lack of stomatal closure, and changes in immune hormone levels, leading to a compromised immune response.

## Data Availability

RNA-Seq data are available at NCBI’s Gene Expression Omnibus GEO Series under accession number GSE236403.

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
