# Peer review of "Arabidopsis Actin-Binding Protein WLIM2A Links PAMP-Triggered Immunity and Cytoskeleton Organization"

_ijms, 2024, doi:10.3390/ijms252111642_

Round 1

Reviewer 1 Report

Comments and Suggestions for Authors

This study explored the function of WLIM2A to link PAMP-triggered immunity, which is meaningful to better understand how the phytohormone(such as actin) regulate MAPK cascades in plant immunity. However, the authors should rewrite the abstract, discussion and conclusion parts with logic and highlight your main meaning of this research. Therefore, I would suggest major revision on the manuscript. In addition, I put forward some suggestions for better modify.

1.    Please rewrite the abstract with logic and highlight your main meaning of this research.

2.    Line14-17: Please use one sentence to introduce the background here.

3.    line 21-28: Please display the main meaning according to your results with logical.

4.    Line 49: Please explain the full name of LIM when it shows at first time.

5.    Line 51: Please explain the full name of SH3 and LD when it shows at first time.

6.    Line 103-113: Please introduce what you are planning to do in this manuscript instead of the results.

7.    Line 116-122 Materials and Methods: Please specify the detailed information about the experiment material, background, and equipment you have used in your experiment in the standard format.

8.    Where are your supplementary figures? Line122 Fig. S2, Line 273 Fig.S3, Line 314 Fig.S4C, Line 386 Fig.S5, Line 507 Fig. S6, etc.

9.    Line 542: What is the (Ref?;FigS7) ? Please have a check.

10.  Discussion and conclusion: The author needs a more in-depth and detailed explanation based on their experiment results with logic order.

11.  Line 603: It is not prospered to put the Figure 7 in this manuscript, because it is a future plan of your work, which is not relevant to this paper. I suggest you would better redraw a workflow figure based on your current work in this manuscript.

Comments on the Quality of English Language

Minor editing of English language required.

Reviewer 2 Report

Comments and Suggestions for Authors

The research investigates the role of the Arabidopsis actin-binding protein WLIM2A in plant immunity, specifically its interaction with mitogen-activated protein kinases (MAPKs) and its impact on the plant's immune response, particularly regarding stomatal immunity and cytoskeleton organization. Strong interaction between WLIM2A and MPK3 was confirmed by BiFC assays and weaker interactions with MPK4 and MPK6. The immunoblotting technique, using the pTEpY antibody, was used to investigate the activation levels of MPK3, MPK4, and MPK6. Indicating higher activation levels of MPK3, MPK4, and MPK6 After flg22 elicitation the wlim2Amutants. Using the CRISPR-Cas9 method, the authors generated WLIM2A knockout lines and conducted a comprehensive analysis of global transcriptomic changes in these mutants. Furthermore, they assessed the plant responses to bacterial and fungal infections, quantified reactive oxygen species production and accumulation levels, and measured changes in salicylic acid (SA) and jasmonic acid (JA) levels in WLIM2A mutants. The findings revealed that WLIM2A knockout lines exhibited compromised responses to the bacterial pathogen Pseudomonas syringae pv. tomato (Pst) but demonstrated enhanced resistance to the fungal pathogen Botrytis cinerea. Additionally, WLIM2A mutants displayed altered stomatal behavior, impacting pathogen entry. Obtained results highlighted differential regulation of immune hormone signaling and biosynthesis genes in WLIM2A mutants, along with enhanced expression of defense-related genes. In conclusion, the authors assert that WLIM2A plays a pivotal role in plant immunity by bridging MAPK signaling to cytoskeleton organization. They emphasize that the phosphorylation of WLIM2A by MAPKs is crucial for regulating stomatal immunity and overall defense responses.

The experiments were carefully planned and the results are described in detail and supported by well prepared figures, which are also adequately described. The results are extensively and carefully discussed in relation to present knowledge, and the conclusions are justified by the results.

I recommend accepting the manuscript in its present form.

Round 2

Reviewer 1 Report

Comments and Suggestions for Authors

Accept in present form